# Paucigranulocytic Asthma: Potential Pathogenetic Mechanisms, Clinical Features and Therapeutic Management

**DOI:** 10.3390/jpm12050850

**Published:** 2022-05-23

**Authors:** Andriana I. Papaioannou, Evangelia Fouka, Polyxeni Ntontsi, Grigoris Stratakos, Spyridon Papiris

**Affiliations:** 12nd Respiratory Medicine Department, “Attikon” University Hospital, Chaidari, 12462 Athens, Greece; xenia-1990@hotmail.com (P.N.); papiris@otenet.gr (S.P.); 2Respiratory Medicine Department, Aristotle University of Thessaloniki, G Papanikolaou Hospital, 57010 Thessaloniki, Greece; evafouka@gmail.com; 31st University Department of Respiratory Medicine, National and Kapodistrian University of Athens, “Sotiria” Hospital, 11527 Athens, Greece; gstratakos@hotmail.com; 4Interventional Pulmonology Unit and ICU of the 1st Respiratory Medicine Department, National and Kapodistrian University of Athens, “Sotiria” Hospital, 11527 Athens, Greece

**Keywords:** asthma, paucigranulocytic asthma, eosinophils, neutrophils, airway inflammation, bronchial thermoplasty

## Abstract

Asthma is a heterogeneous disease usually characterized by chronic airway inflammation, in which several phenotypes have been described, related to the age of onset, symptoms, inflammatory characteristics and treatment response. The identification of the inflammatory phenotype in asthma is very useful, since it allows for both the recognition of the asthmatic triggering factor as well as the optimization of treatment The paucigranulocytic phenotype of asthma (PGA) is characterized by sputum eosinophil levels <1–3% and sputum neutrophil levels < 60%. The precise characteristics and the pathobiology of PGA are not fully understood, and, in some cases, it seems to represent a previous eosinophilic phenotype with a good response to anti-inflammatory treatment. However, many patients with PGA remain uncontrolled and experience asthmatic symptoms and exacerbations, irrespective of the low grade of airway inflammation. This observation leads to the hypothesis that PGA might also be either a special phenotype driven by different kinds of cells, such as macrophages or mast cells, or a non-inflammatory phenotype with a low grade of eosinophilic inflammation. In this review, we aim to describe the special characteristics of PGA and the potential therapeutic interventions that could be offered to these patients.

## 1. Introduction

Asthma is a heterogeneous disease usually characterized by chronic airway inflammation, in which several phenotypes have been described, related to the age of onset, symptoms, inflammatory characteristics and treatment response [1,2]. Using sputum analysis to identify the pattern of airway inflammation, four different inflammatory subtypes can be recognized [3]: the neutrophilic phenotype, in which subjects have a sputum neutrophil proportion ≥ 61%; the eosinophilic phenotype, in which eosinophils occur in a proportion of ≥1–3% [3,4,5]; the mixed granulocytic phenotype, in which both neutrophils and eosinophils occur above the aforementioned levels; and, finally, the paucigranulocytic phenotype, in which both types of cells are below the aforementioned levels [3].

The identification of the inflammatory phenotype in asthma is very useful, since it allows for both the recognition of the asthmatic triggering factor as well as the optimization of treatment. Briefly, sputum eosinophilia is often related to the presence of atopy, and, most of the time, it is characterized by a good response to corticosteroids [6,7]. On the other hand, the presence of increased neutrophil numbers is usually related to exposure to cigarette smoke or other air pollutants or to the presence of infection, either bacterial or viral [8]. However, it is important to point out that most of these studies are mainly cross-sectional, and data are obtained on steroid-treated patients and not only on steroid-naïve patients, making it difficult to conclude whether paucigranulocytic asthma is a phenotype of treatment effectiveness or a separate phenotype.

Although the precise characteristics and the pathobiology of paucigranulocytic asthma (PGA) are not fully understood, in many cases, it seems to represent a phenotype of a previous eosinophilic asthma with a good response to anti-inflammatory treatment [9]. However, several studies have shown that it might also be either a special phenotype driven by different kinds of cells, such as macrophages or mast cells [10,11], or a non-inflammatory phenotype with a low grade of eosinophilic inflammation [12]. In such a case, these special characteristics may also probably be the reason that PGA shows minimal or no response to therapy.

PGA is one of the most common asthma phenotypes in optimally treated asthmatics, with an estimated incidence between 31 and 47% of the total asthmatic population [9,12,13]. However, although in the great majority of these patients, asthma is characterized as “benign”, with a good response to corticosteroids—which probably explains the disappearance of eosinophilic inflammation—approximately 10–15% of patients with PGA, representing 5–10% of treated asthmatics, have poor asthma control and are still at an increased risk for asthma exacerbations [9,14]. Proposed clinical phenotypes of paucigranulocytic (PGA) asthma are shown on Figure 1.

## 2. Underlying Mechanisms of PGA

### 2.1. Low Intensity of Inflammation

Although the exact pathogenesis of PGA remains unexplored, previous studies support the hypothesis that PGA represents a phenotype with a low grade of inflammation compared to the other three phenotypes (eosinophilic, mixed, neutrophilic) [9,12]. However, the underlying inflammatory process seems to be of greater intensity compared to healthy controls [12]. Furthermore, in PGA, the expression of inflammatory mediators such as the active-matrix metalloproteinase-9 (MMP-9) and neutrophil elastase do not differ from non-asthmatic controls, and both are lower compared to neutrophilic asthma [15], whereas BAL levels of IL-17, IL-6, G-CSF, IL-12, TNF-α and IFN-γ were enhanced in neutrophilic cohorts compared to PGA [16]. Similarly, sputum levels of IL-1β and IL-8 have been found to be lower in patients with PGA compared to neutrophilic asthma [9,17,18], and eosinophilic cationic protein (ECP) levels in sputum were lower in PGA compared to patients with eosinophilic asthma [9].

PGA seems to be mainly characterized by structural cell abnormalities, which seem to play a key role in the pathogenesis of this condition, especially when asthma control is poor. Intrinsic abnormalities in the function of structural cells (such as epithelial cells, smooth muscle cells, vessels and neuronic cells) seem to be pivotal, and these cells seem to orchestrate and preserve airway inflammation [19] by secreting several paracrine or autocrine factors, which results in the modulation of cellular functions such as agonist-induced shortening, secretion and migration [20]. Li et al. have demonstrated a different gene expression pattern in PGA with moderate discrimination ability, suggesting a decreased immune cell infiltration and function and further supporting the low intensity of the inflammatory process in PGA [21]. In this study, the genes most relevant to PGA that were detected were *ADCY2*, *CXCL1*, *FPRL1*, *GPR109B*, *GPR109A* and *ADCY3*, with earlier studies suggesting that *ADCY3* upregulation is related to the suppressed function of dendritic cells [22], whereas the down-regulation of the other five genes has been shown to decrease chemotaxis and immune cell migration to the inflammatory sites [23,24,25].

However, studies that used human airway smooth muscle (ASM) cell cultures derived from subjects with fatal asthma have shown a greater proliferation of growth factors compared with age- and sex-matched controls [26,27], and it has also been shown that repair mechanisms in airway epithelium cells from asthmatic patients are dysfunctional [28]. Both epithelial cells and ASM secrete several cytokines (such as TNF-α, IFN-1, IFN-2, IL-6 and IL8), chemokines and adhesion molecule 1 (ICAM-1), as well as prostaglandins and eicosanoids that are steroid-insensitive and potentially modulate airway function [29,30,31,32,33].

### 2.2. Uncoupling of Airway Remodeling and Hyperresponsiveness from Inflammation

Although it has a low inflammatory profile, in some cases, PGA is an asthmatic phenotype also related to severe refractory asthma and poor asthma control [9]. This observation leads to the conclusion that, in these patients with PGA, there is also an uncoupling of the airway obstruction from airway inflammation. This distinction is probably driven by neuronal factors, non-immunological mediators, signaling molecules and susceptible genes, which all change ASM contractility and predispose one to airway hyperresponsiveness (AHR), irrespective of the inflammatory status [17].

Several animal studies have supported the uncoupling of AHR and obstruction from airway inflammation in the pathogenesis of PGA [17], providing evidence that processes evoking airway hyperresponsiveness and ASM thickening occur independently from inflammation and may be a consequence of a loss of negative homeostatic processes. Although there are studies supporting the neuro-immune link [34], in mice treated with nerve growth factor (NGF), AHR was induced at the same degree as in allergen-sensitized mice without an induction of inflammation [35]. Moreover, the dysregulation of critical signing molecules, such as G protein-coupled receptors, transmembrane proteins and growth transcriptional factors, is considered to be an additional possible mechanism of promoting AHR independently of inflammation [36,37,38]. In other animal studies, an overexpression of asthma genes located in chromosome 17q21 (including gasdermin B (*GSDMB*) and human orosomucoid-like 3 (*ORMDL3*)) have been detected. It is known that the upregulation of these genes leads to airway remodeling, AHR, increase in ASM mass, subepithelial fibrosis and mucous production in the absence of airway inflammation [39,40].

Similarly, in human studies, functional (such as increased contractility) and structural (such as hypertrophy and thickening) changes of ASM have been observed in patients with airway hyperresponsiveness [41]. Elliot et al. used lung tissue obtained from subjects derived from five post-mortem studies of asthma and concluded that, in those with PGA, there was only a thickness of the ASM layer and basement membrane, which was considered to be inflammation-independent. On the contrary, in cases of granulocytic asthma, there was also an increase in the inner and outer airway wall thickness and a narrowing of the airway lumen due to ASM shortening and mucus obstruction, an observation that suggests a coexisting inflammatory process [42].

## 3. Clinical Features of PGA

Based on induced sputum analysis, PGA seems to be the most common inflammatory asthma phenotype in steady state in several clinical trials [9,43]. However, the association of PGA existence with asthma severity remains unclear. Since PGA is characterized by the absence of increased inflammatory cells in the airways, it might be considered to be the result of a successful therapeutic intervention in the absence of neutrophilic inflammation. In fact, the majority of PGA patients maintained better lung function based on post-bronchodilation FEV1 (% predicted) and FEV1/FVC ratio as compared to patients with other asthma phenotypes, whereas severe refractory asthma criteria were met less often [9]. However, despite these findings, a significant proportion of patients with PGA (21.7%) were characterized as having severe refractory asthma, whereas 14.8% of patients with PGA had poor asthma control as assessed by an Asthma Control Test (ACT) score ≤ 19 [9]. This observation leads to the conclusion that both airway remodeling and AHR could be present in PGA irrespective of inflammation.

In a recent study, Deng et al. prospectively examined a cohort of 145 patients with PGA and were able to identify three clusters [14]: Cluster 1, characterized as ‘mild PGA’, was the most common (75.9%), and included mainly female non-smoker patients with nearly normal lung function, indicating mild airway obstruction and good asthma-related quality of life. Cluster 2 included patients with atopy, psychological dysfunction and rhino-conjunctivitis. Cluster 3 was smoking-associated and characterized by older age, later asthma onset and significantly increased risk of severe exacerbations requiring emergency visits and hospitalization [14]. This cluster analysis shows that PGA might not represent a single entity, but could also be the result of different inflammatory processes, resulting in specific structural, inflammatory and functional manifestations in combination with the effects of therapy.

## 4. Clinical Management of PGA

Corticosteroids are the basis of asthma controller therapy; however, non-T2 asthma is typically corticosteroid-resistant [44]. Moreover, no other specific therapies, such as long-acting muscarinic antagonists (LAMAs), beta-2 adrenergic agonists oral macrolides and the currently available biologic therapies, have shown any clinical benefits in patients with asthma that is associated with a non-T2 inflammatory process [45]. Other available treatment options for non-T2 asthma (i.e., trigger avoidance, vaccination against respiratory pathogens, smoking cessation and weight reduction in obese asthmatics) are not mechanism-based, but there is some evidence suggesting that non-T2 asthma patients with AHR may be potential candidates for bronchial thermoplasty [46]. As a heterogeneous disease with diverse underlying endotypes, a precise characterization of T2-low asthma may be necessary in order to develop effectively tailored treatments.

## 5. Pharmacological Therapies

### 5.1. Inhaled Corticosteroids (ICS)

Although anti-inflammatory therapies result in significant improvements in eosinonophilic asthma, non-T2 asthma is persistently corticosteroid-resistant [47]. Ntontsi and colleagues have provided strong evidence that non-T2 asthma (which includes both neutrophilic and PGA endotypes) does not benefit from corticosteroid therapy, despite optimal dosing and adherence [9]. Similarly, Demarche et al. did not find any statistically significant difference in sputum cell counts between PGA patients who were treated and those who were not treated with ICS [12]. In conclusion, the evidence suggests that PGA is a potentially corticosteroid-insensitive phenotype, and alternative therapeutic interventions are required.

### 5.2. Macrolides

Macrolides have a well-known important clinical efficacy in treating a wide variety of pulmonary disorders, including refractory asthma, due to their well-established antimicrobial and anti-inflammatory properties [48]. In an early study, Simpson et al. showed that long-term clarithromycin therapy resulted in reductions in sputum IL-8 levels and neutrophil counts in patients with severe refractory asthma; however, no benefit in clinical outcomes such as lung function, dose response to hypertonic saline or asthma control scores was demonstrated [49]. Similarly, another study that assessed the efficacy of long-term azithromycin administration in smokers with asthma, an understudied patient group, demonstrated its lack of efficacy in both clinical and laboratory outcomes. On the other hand, in two large, randomized control trials, azithromycin significantly improved asthma exacerbations and quality of life in both eosinophilic and noneosinophilic asthma. In the Azithromycin for Prevention of Exacerbations in Severe Asthma Trial (AZISAST), add-on azithromycin (250 mg daily three times per week) resulted in significantly fewer severe asthma exacerbations during the 26-week study period compared to a placebo in a pre-defined subgroup analysis of severe non-eosinophilic asthma patients with a history of severe exacerbations [50]. Similarly, in the Asthma and Macrolides: Azithromycin Efficacy and Safety (AMAZES) trial, azithromycin 500 mg three times per week significantly reduced the rate of moderate and severe exacerbations and improved asthma-related quality of life after 52 weeks of therapy in both the eosinophilic and noneosinophilic asthma subgroups of patients [51]. In a subsequent study, Taylor et al. assessed the effect of azithromycin on sputum microbiology in participants of the AMAZES study and concluded that azithromycin efficacy was unlikely to be a result of its antibiotic effect, as they found that azithromycin did not affect the total airway bacterial load compared to a placebo [52]. Based on current evidence, the European Respiratory Society/American Thoracic Society severe asthma guidelines recommend the use of azithromycin in severe asthma that does not respond to treatment [53]. More studies are needed to examine the potential immunomodulatory effects of long-term, low-dose macrolide with a focus in patients with PGA asthma; however, in our opinion, a therapeutic trial is at least justified for these patients, due to the lack of specific treatment options.

### 5.3. LAMAs

The Global Initiative for Asthma (GINA) recommendations have recently been expanded to include triple combination ICS-LABA-LAMA for patients in Step 5 if their asthma is persistently uncontrolled despite medium- or high-dose ICS-LABA use [54]. In early studies, tiotropium demonstrated clinical efficacy in patients with fixed airway obstruction, an asthma phenotype usually associated with neutrophilic airway inflammation [55], and in patients with T2-low asthma [56]. Casale et al., in an exploratory analysis of clinical trials of tiotropium for asthma, reported that the clinical benefit of tiotropium was independent of the T2 phenotype, as it was stated by total IgE and blood eosinophils levels [57]. Likewise, in a recently published study, the addition of umeclidinium to an ICS/LABA combination resulted in significant, yet small, dose-related improvements compared with ICS/LABA alone, irrespective of the baseline blood eosinophil counts and exhaled nitric oxide levels [58]. 

## 6. Non-Pharmacological Strategies

### 6.1. Smoking Cessation

In asthmatic patients who smoke, disease control is poorer and lung function decline is faster than in nonsmoking asthmatics [59]. Cigarette smoking may modify inflammation that is associated with asthma, and evidence shows both heightened and suppressed inflammatory responses in smokers, compared with nonsmokers with asthma [60]. Reduced responsiveness to corticosteroid therapy is a major barrier to the effective management of smoking asthmatics [61], and potential mechanisms of corticosteroid resistance may include alterations in inflammatory cell phenotypes in asthmatic airways (e.g., increased neutrophils or reduced eosinophils) [62], changes in the glucocorticoid receptor alpha-to-beta ratio [63,64], an increased activation of pro-inflammatory transcription factors [65] or reduced histone deacetylase activity [66]. Thus, smoking cessation and the avoidance of exposure to second-hand smoke and environmental/occupational pollutants are important and low-cost, but often disregarded, interventions [67]. Several studies have demonstrated that smoking cessation was associated with the reduction of neutrophilic inflammation and exacerbations and significant improvements in symptoms, lung function and asthma control, compared to those asthmatics who continued to smoke [68,69,70]. Therefore, smoking cessation strategies should be incorporated in the clinical management of every patient.

### 6.2. Weight Reduction

Current evidence suggests that obesity may increase the risk of developing asthma, with underlying pathobiological mechanisms influencing both type 1 and type 2 inflammation [71,72]. Indeed, in obese asthmatics, sputum cell counts were reported to be normal as compared to non-obese patients [73], and the vast majority of evidence suggests that symptoms in obese patients may be related to underlying AHR [74,75], mechanical factors related to increase expiratory reserve volumes [76,77], smooth muscle actin-myosin cross-linking [78,79], adipose tissue accumulation [80,81] and other metabolic or insulin resistance mechanisms [82], rather than to an inflammatory process. On the other hand, asthma may increase the risk for obesity, due to reduced physical activity and oral corticosteroid use [83]. In a randomized study involving obese patients with severe asthma, weight loss was associated with significant improvements in symptoms, lung function and asthma control, although no significant changes in sputum inflammatory phenotypes were demonstrated [84]. Bariatric surgery has also been shown to significantly reduce systemic inflammatory markers, including high-sensitivity C-reactive protein and leptin, in patients with and without asthma, although there was no change in submucosal eosinophil, neutrophil and macrophage cell counts [85]. In another large self-controlled case series of obese patients with asthma, bariatric surgery resulted in reductions in emergency department visits and asthma-related hospitalizations in the first year post-surgery [86]. It is likely that, in addition to immunologic, metabolic and cardiovascular benefits, weight loss will result in better asthma control, a fact that may be attributable to improvements in chronic deconditioning, mechanical restriction and potentially comorbid sleep apnoea [87]. Regardless of the underlying mechanism, it is reasonable to consider bariatric surgery or other effective weight reduction strategies in asthmatics with morbid obesity.

### 6.3. Bronchial Thermoplasty

Bronchial thermoplasty (BT) is an endoscopic treatment for persistent, uncontrolled asthma, which involves the delivery of radio frequency energy to the airways to reduce airway smooth muscle mass [46]. It seems that BT decreases ASM mass and nerve fibres in epithelium, and, thus, the attenuation of neural reflexes by locally delivered thermal energy is probably one of the mechanisms by which this intervention is effective; however, alterations in airway epithelial, gland and/or nerve function improvements in small airway function or a placebo effect may also be relevant [88,89]. Papakonstantinou et al. evaluated the histopathological alterations collected by endobronchial biopsies before and after three sequential BTs in 30 patients with severe uncontrolled asthma and suggested that these appear to be distinct in different endotypes/phenotypes, indicating that these changes are probably achieved by diverse molecular targets [90]. Several studies have demonstrated that BT results in improvements in lung function, asthma control and quality of life and increases in symptom-free days, as well as a decrease in rescue medication use [91,92]. In the AIR2 trial [90], BT resulted in significant reductions in severe exacerbation rates, emergency department visits and days missed from work/school. However, a substantial sham effect was described in this study, including a clinically meaningful improvement in AQLQ score (≥0.5) in 64% of the sham group versus 79% in the BT group, a finding discussed by the authors as resulting from preconceived expectations and the care and attention provided by the study staff [93]. Recently, the randomized controlled TASMA study [94] reported a significant decrease of ASM mass in asthmatic patients undergoing bronchial biopsy before and after BT when compared with a randomized non-BT-treated control group, even though it failed to show a correlation of this finding with clinical outcomes. The same effect of ASM mass and extracellular matrix reduction following a BT procedure, representing changes in the remodeling of asthmatic airways, has consistently been shown with the less invasive Optical Coherence Tomography bronchoscopic imaging technique not requiring biopsy [95]. Moreover, an international, multicentre study of asthmatic patients who were previously enrolled in the AIR [91], RISA [92] and AIR2 [93] studies has shown that the beneficial effects of BT, in terms of a reduction in exacerbation rate and an improvement in quality of life, are sustained for 10 years or more, with an acceptable safety profile [96].

Based on the limited evidence for long-term efficacy and safety, international guidelines do not currently recommend BT as a routine procedure in the management of severe asthma [53]. The European Respiratory Society/American Thoracic Society guidelines for severe asthma management [53] recommend that BT should be performed in the context of an Institutional Review Board-approved independent systematic registry or a clinical study. In the same way, the 2021 GINA guidelines [54] suggest that BT may be considered as a treatment option for adult patients at Step 5, whose asthma remains uncontrolled despite an optimization of asthma therapy, after referral to a severe asthma specialty centre and only in the context of an independent Institutional Review Board-approved systematic registry or clinical study.

## 7. Future Therapeutic Targets

Cysteine-X-cysteine (CXC) chemokine receptors bind IL-8 and play an important role in neutrophils migration to inflammatory sites [97]. In a large, multicentre, dose-titrating trial, O’Byrne et al. [98] investigated AZD5069, a selective CXCR2 antagonist, in patients with uncontrolled persistent asthma despite a medium-to-high dose of ICS and LABA, and they reported no significant benefits in rates of severe exacerbations, lung function and ACQ-5 scores. As sputum differentials identified by sputum induction techniques were not used in this study, the proportion of patients who definitely had neutrophilic asthma was uncertain; therefore, these results may not be limited only to patients with neutrophilic asthma.

Thymic Stromal Lymphopoeitin (TSLP), an alarmin released by airway epithelial cells after stimulation by a variety of irritants, such as airway pollutants, smoke, infections or allergens [99], impacts both T2 and non-T2 pathways; thus, an inhibition of this molecule may present a novel therapeutic target for T2-low asthma [100]. Corren et al. [101], in a phase 2, randomized, double-blinded, placebo-controlled, dose-ranging trial of the anti-TSLP antibody tezepelumab, reported a similar rate of clinically significant exacerbations in adult patients with moderate-to-severe uncontrolled asthma that received tezepelumab compared to those who received a placebo. In a subpopulation analysis of the annualized exacerbation rate, the primary outcome of the study, the benefit from tezepelumab, was independent of baseline blood eosinophil counts, a finding that suggests that this anti-TSLP antibody may also be effective for T2-low asthma. The phase 3 NAVIGATOR trial [102] assessed the efficacy of tezepelumab, administered subcutaneously every 4 weeks for 52 weeks, versus a placebo in patients with severe uncontrolled asthma. In a pre-specified subgroup analysis, according to baseline FeNO level, perennial-allergy status, total serum IgE level and blood eosinophil counts, the annualized rate of asthma exacerbations was significantly reduced in patients with both high and low blood eosinophil counts (<300 cells per microliter) at baseline. In this study, significant improvements have also been reported in other secondary outcomes (FEV1 and scores on the ACQ-6, AQLQ(S)+12 and ASD); however, the greatest benefits of tezepelumab were demonstrated in those patients with a blood eosinophil count of at least 300 cells per microliter. Conclusively, the results of these preliminary clinical trials suggest that TSLP is a promising therapeutic target for at least some patients with T2-low asthma.

In a randomized controlled trial of patients with uncontrolled moderate-to-severe asthma on inhaled corticosteroids, Brodalumab, a human monoclonal antibody that blocks the activity of IL-17A, IL-17B and IL-25 through binding to IL-17RA, failed to show a change in ACQ score, asthma symptoms or lung function in the whole population; however, the authors reported a trend towards improvement in ACQ score in a small subgroup of patients with high bronchodilator reversibility [103]. Unfortunately, the unfavorable safety profile led to a discontinuation of the further development of this agent for asthma. The results of a phase 2 randomized clinical trial of another anti-IL17A antibody, secukinumab, in patients with inadequately controlled moderate-to-severe T2-low asthma have not yet been published [104]. The clinical management of paucigranulocytic asthma is shown in Figure 2.

## 8. Conclusions

Our lack of understanding of the pathogenesis, therapy and prognosis of PGA asthma represents a profound unmet need. PGA is a heterogeneous disease with diverse endotypes; therefore, it is of cardinal importance to unveil the underlying inflammatory and immune pathogenetic mechanisms. The lack of efficacy of glucocorticosteroids in the modulation of structural cell function highlights the necessity for novel therapeutic targets for pauci-immune asthma. Since surrogate inflammatory markers are present in the absence of any granulocytic infiltration, other features, such as the inflammatory mediators generated by airway structural cells, should be investigated; however, existing challenges in characterizing structural cell function in asthma have impeded research progress. Improved noninvasive imaging may provide opportunities for deep phenotype patients and for devising and predicting responses to new therapeutic agents. International collaborative programs, such as the Unbiased Biomarkers for the Prediction of Respiratory Disease Outcome (U-BIOPRED) study [105] and the UK Refractory Asthma Stratification Programme (RASP-UK) [106] aim to identify new phenotypes/endotypes and treatment targets for future research for patients with paucigranulocytic asthma.

## Figures and Tables

**Figure 1 jpm-12-00850-f001:**
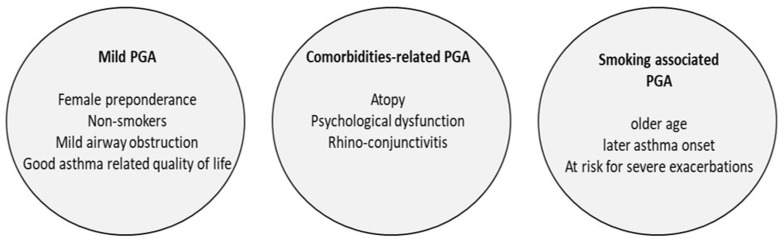
Proposed clinical phenotypes of paucigranulocytic (PGA) asthma (adapted from Deng et al.) [14].

**Figure 2 jpm-12-00850-f002:**
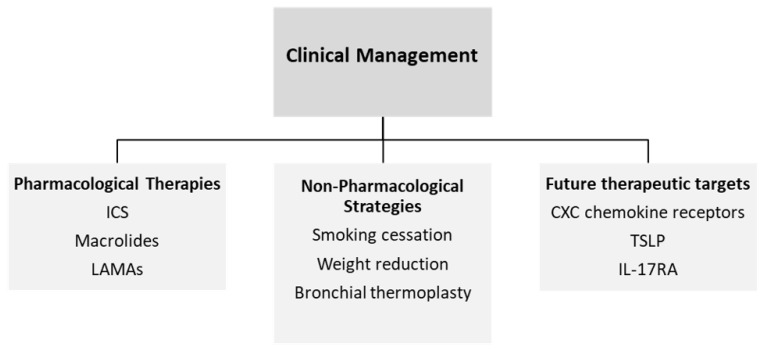
Clinical management of paucigranulocytic asthma.

## Data Availability

Not applicable.

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
