# Peer review of "Paucigranulocytic Asthma: Potential Pathogenetic Mechanisms, Clinical Features and Therapeutic Management"

_jpm, 2022, doi:10.3390/jpm12050850_

Round 1

Reviewer 1 Report

The manuscript provides a comprehensive overview on one of the inflammatory subgroup of asthma, the paucigranulocytic phenotype. The authors put together a good structure for different aspects covering different aspects. References are used properly overall.

I have some comments to improve the clarity of the mansucript:

The authors define asthma as „heterogenous inflammatory disease” in the first line of the abstract and also at other parts. This term does not correlate with the current definition of asthma in GINA or other current guidelines that define asthmaas a heterogeneous disease, usually characterised by chronic airway inflammation. Furthermore they decribe paucigranulocytic inflammation as a subtype of non-allergic asthma among the clinical phenotypes. I fully recommend to change the text and adhere to the current definition. That helps the appropriate framing of the subgroup the manuscript deals with.  

In the introduction it would be useful to clarify that studies investigating inflammatory phenotypes are mainly cross-secsional and data obtained on steroid-treated patients and not only on steroid-naive ones to address the dilemma of a treatmnet effect phenotype or a separate phenotype better. This aspect is even more important when underlying mechanisms are addressed.

 Regarding the treatment session, I suggest to concentrate it more to PGA and shorten it appropriately.

I would welcome some figures to support understanding.

Regarding the reference list, my suggestion is to add some papers on asthma clusters, and mechanistic papers such as

Clinically relevant subgroups of COPD and asthma by Turner AM et al in Eur Respir Review;

Mechanisms of non-type2 asthma by Hudey Sm et al in Current Opinion Immunol;

A paucigranulocytic asthma host environment promotes the emergence of virulent influenza viral variants by Hulme KD et al in eLife

Bronchoaleveolar cytokine pattern in children… by Steinke JW JACI.

Author Response

Reviewer 1

The manuscript provides a comprehensive overview on one of the inflammatory subgroup of asthma, the paucigranulocytic phenotype. The authors put together a good structure for different aspects covering different aspects. References are used properly overall.

I have some comments to improve the clarity of the manuscript:

The authors define asthma as „heterogenous inflammatory disease” in the first line of the abstract and also at other parts. This term does not correlate with the current definition of asthma in GINA or other current guidelines that define asthma as a heterogeneous disease, usually characterized by chronic airway inflammation. Furthermore, they describe paucigranulocytic inflammation as a subtype of non-allergic asthma among the clinical phenotypes. I fully recommend changing the text and adhere to the current definition. That helps the appropriate framing of the subgroup the manuscript deals with.  

We thank the reviewer for his/her comment.

The definition has been altered as follows:

“Asthma is a heterogeneous disease, usually characterized by chronic airway inflammation in which several phenotypes have been described, related to age of onset, symptoms, inflammatory characteristics and treatment response”.

Regarding the definition of paucigranulocytic inflammation, the following definitions appear in the text

“Although the precise characteristics and the pathobiology of paucigranulocytic asthma (PGA) are not fully understood, in many cases it seems to represent a phenotype of a previous eosinophilic asthma with a good response to anti-inflammatory treatment [8]. However, several studies have shown that it might also be either a special phenotype driven by different kinds of cells, such as macrophages or mast cells [9, 10], or a non-inflammatory phenotype, with a low grade of eosinophilic inflammation [11].”

In the introduction it would be useful to clarify that studies investigating inflammatory phenotypes are mainly cross-sectional and data obtained on steroid-treated patients and not only on steroid-naive ones to address the dilemma of a treatment effect phenotype or a separate phenotype better. This aspect is even more important when underlying mechanisms are addressed.

We thank the reviewer for his/her comment. A phrase has been added in the introduction section as follows

“However, it is important to point that most of these studies are mainly cross-sectional and data obtained on steroid-treated patients and not only on steroid-naïve making difficult to conclude on whether paucigranulocytic asthma is a treatment effect phenotype or a separate phenotype.”

Regarding the treatment session, I suggest to concentrate it more to PGA and shorten it appropriately.

We thank the reviewer for his/her comments. We agree that macrolides are not a specific therapy for PGA. However, we believe that this treatment option should be tried in patients who do not respond to other treatments especially when there is lack of other treatment options.

For this reason, a comment has been added in the end of the paragraph as follows:

“More studies are needed to examine the potential immunomodulatory effects of long-term low‐dose macrolide with focus in patients with PGA asthma, however, in our opinion, a therapeutic trial is at least justified for these patients, due to the lack of specific treatment options.”

However, in case that the reviewer insists to cut off the treatment section, we will be more than happy to do so.

I would welcome some figures to support understanding.

We thank the reviewer for his/her comment. Two figures have been included in the manuscript and now appear as Figure 1 and 2

Regarding the reference list, my suggestion is to add some papers on asthma clusters, and mechanistic papers such as

Clinically relevant subgroups of COPD and asthma by Turner AM et al in Eur Respir Review;

Mechanisms of non-type2 asthma by Hudey Sm et al in Current Opinion Immunol;

A paucigranulocytic asthma host environment promotes the emergence of virulent influenza viral variants by Hulme KD et al in eLife

Bronchoaleveolar cytokine pattern in children… by Steinke JW JACI.

We thank the reviewer for his/her comment.

Three of the references have been added in the text

Reviewer 2 Report

I have reviewed the manuscript with great interest.

The main purpose of the research was the introduction of the inflammatory subtype of paucigranulomatisis asthma (PGA) The main question was to analyze pathobiology, mechanisms and to discuss possible properly tailored treatment of this type of asthma. This purpose by authors was obtained with success. It will be worth to describe more examples and characteristics of three mentioned cluster of PGA . Authors have mentioned three clinical features of PGA but the precise characterization of mechanisms is much more problematic. There are three clusters; first mild asthma in non-smoking women with normal or mild airway obstruction, the second in atopic patients with associated rhinoconjunctivities and the third later asthma onset in smoking, older patients. From practical point of view it is very difficult to differentiate subtypes using induced sputum and we must remember about possible overlap of the different subtypes of asthma. Thus Authors suppressed the lack of proper marker of PGA. In the aim to deep phenotype new non- invasive methods are needed. The text is clear well written and inspirating. The conclusions are consistent with the evidences and arguments.

Author Response

Reviewer 2

I have reviewed the manuscript with great interest.

The main purpose of the research was the introduction of the inflammatory subtype of paucigranulomatisis asthma (PGA) The main question was to analyze pathobiology, mechanisms and to discuss possible properly tailored treatment of this type of asthma. This purpose by authors was obtained with success. It will be worth to describe more examples and characteristics of three mentioned cluster of PGA. Authors have mentioned three clinical features of PGA but the precise characterization of mechanisms is much more problematic. There are three clusters; first mild asthma in non-smoking women with normal or mild airway obstruction, the second in atopic patients with associated rhinoconjunctivities and the third later asthma onset in smoking, older patients. From practical point of view, it is very difficult to differentiate subtypes using induced sputum and we must remember about possible overlap of the different subtypes of asthma. Thus, Authors suppressed the lack of proper marker of PGA. In the aim to deep phenotype new non- invasive methods are needed. The text is clear well written and inspirating. The conclusions are consistent with the evidences and arguments.

We thank the reviewer for his/her comments